# Metabolic Contribution and Cerebral Blood Flow Regulation by Astrocytes in the Neurovascular Unit

**DOI:** 10.3390/cells11050813

**Published:** 2022-02-25

**Authors:** Shinichi Takahashi

**Affiliations:** 1Department of Neurology and Stroke, Saitama Medical University International Medical Center, 1397-1 Yamane, Hidaka-shi 350-1298, Japan; takashin@tka.att.ne.jp; Tel.: +81-42-984-4111 (ext. 7412) or +81-3-3353-1211 (ext. 62613); Fax: +81-42-984-0664 or +81-3-3357-5445; 2Department of Physiology, Keio University School of Medicine, 35 Shinanomachi, Shinjuku-ku, Tokyo 160-8582, Japan

**Keywords:** astrocyte, astroglia, astrocyte-neuron lactate shuttle, glucose, lactate, functional hyperemia

## Abstract

The neurovascular unit (NVU) is a conceptual framework that has been proposed to better explain the relationships between the neural cells and blood vessels in the human brain, focused mainly on the brain gray matter. The major components of the NVU are the neurons, astrocytes (astroglia), microvessels, pericytes, and microglia. In addition, we believe that oligodendrocytes should also be included as an indispensable component of the NVU in the white matter. Of all these components, astrocytes in particular have attracted the interest of researchers because of their unique anatomical location; these cells are interposed between the neurons and the microvessels of the brain. Their location suggests that astrocytes might regulate the cerebral blood flow (CBF) in response to neuronal activity, so as to ensure an adequate supply of glucose and oxygen to meet the metabolic demands of the neurons. In fact, the adult human brain, which accounts for only 2% of the entire body weight, consumes approximately 20–25% of the total amount of glucose and oxygen consumed by the whole body. The brain needs a continuous supply of these essential energy sources through the CBF, because there are practically no stores of glucose or oxygen in the brain; both acute and chronic cessation of CBF can adversely affect brain functions. In addition, another important putative function of the NVU is the elimination of heat and waste materials produced by neuronal activity. Recent evidence suggests that astrocytes play pivotal roles not only in supplying glucose, but also fatty acids and amino acids to neurons. Loss of astrocytic support can be expected to lead to malfunction of the NVU as a whole, which underlies numerous neurological disorders. In this review, we shall focus on historical and recent findings with regard to the metabolic contributions of astrocytes in the NVU.

## 1. Introduction

The neurovascular unit (NVU) is a conceptual framework proposed to explain brain functions in health and disease in terms of the interactions between the component cells and the microcirculation in the brain [1,2,3,4,5] (Figure 1). The concept of the NVU was first proposed in 2002 [6] with the goal of developing innovative treatment strategies for ischemic stroke. Since then, it has been used as a tool not only to develop treatments for acute ischemic stroke [7,8], but also to develop therapies for slowly progressive neurodegenerative diseases characterized by abnormal accumulation of proteins [9,10,11,12], as well as for autoimmune neurological diseases linked to inflammation in the body and brain [13,14]. Understanding how the microvasculature interacts with various types of neural cells under normal physiological conditions is essential to obtaining an understanding of the pathogenesis and pathophysiology of numerous neurological disorders [7,8,9,10,11,12,13,14,15,16,17,18].

The core function of the normal brain is information processing by neurons, which is based on the movements of multiple ions across the cells. In particular, the generation of action potentials, which result from influx and efflux of Na^+^ and K^+^ across the neuronal cell membrane, forms the basis of information transmission by neurons to distant sites. In order to maintain the Na^+^ and K^+^ concentration gradient and facilitate ion transport against the gradient, the neurons require energy in the form of adenosine triphosphate (ATP) [19,20,21]. In fact, the ATP consumption by Na^+^,K^+^-ATPase in the basal state accounts for approximately 50% of the total ATP consumption of the brain. In the adult brain, ATP production is exclusively a product of oxidative metabolism of glucose [22,23]. Since the brain contains neither glucose nor oxygen stores, both glucose and oxygen must be transported from outside the brain by the cerebral blood flow (CBF). The CBF is precisely regulated at all levels of brain vessels, from the main arterial trunks to the small arteries as well as capillaries [1,2,3,4,5,7,8]. The endothelial cells and astrocytes play important roles in the exchange of substances between the blood and neurons, because the end feet of the astrocytes almost completely ensheath the surfaces of the capillaries [19,20,21]. Alterations of the CBF occur in response to changes in neuronal activities, and changes in vascular caliber are observed even at the capillary level [1,2,3,4,5,7,8]. Regulation of the CBF is mainly mediated by vascular smooth muscle contraction and relaxation at the level of the upstream arterioles. However, it has been pointed out that pericytes may be involved in the dilatation and contraction of microvessels even at the capillary level [24,25,26,27,28]. In addition to pericyte regulation, the capillary bed size may also be actively regulated by an increase or decrease in water content in the Virchow–Robin space [29,30,31], the space between the astrocytic end feet and the outer surfaces of the microvessels. Since the astrocyte-ensheathed capillaries are not in direct contact with the neurons, neurons may actually receive the required oxygen and glucose for ATP production for Na^+^,K^+^-ATPase activity from astrocytes, probably in the form of a partial metabolite (i.e., lactate or ketone bodies) [19,20,21]. In addition, oxygen and glucose diffusing into the extracellular fluid may also be taken up directly by the neurons.

## 2. Construction of the Virtual Concept of an NVU

The term NVU in its current sense was first used by the Stroke Progress Review Group of the National Institute of Neurological Disorders and Stroke (NINDS) [6]. In 2001, the Director of NINDS, Gerald Fischbach, convened a group of basic and clinical researchers on cerebrovascular disease in the United States to address the need for a comprehensive understanding of the pathogenesis of stroke and to translate it to clinical application. A report was published one year later, in 2002, under the co-chairs of James C. Grotta and Michael A. Moskowitz. The members of the group included Costantino Iadecola and Gregory J. del Zoppo, who are leaders in this area of research. The report lists 15 priority areas for advancing research in cerebrovascular disease, and the second of the five goals common to all of them is “understanding the complex functions of the blood per se, vessel walls, and the brain”. The term NVU, components of which are the blood vessel wall, extracellular matrix (ECM), neurons, and glia, was coined by this group. In addition, the 4th of the 15 priority areas, namely, “Neurovascular Protective Mechanisms”, which refers to the endothelial cells and astrocytes, also involves the NVU. The NVU is defined as the sum of the endothelial cells, astrocytes and pericytes as cellular elements, and the ECM as a non-cellular element, and also the proteins and their regulatory enzymes related to these components. The group has called special attention to the roles of the “glio-vascular module” within the NVU [5,6]. However, it is generally accepted that the NVU is a structural unit that includes both the glio-vascular module and neurons. In addition, immune and inflammatory cells, such as microglia [32,33,34,35] and oligodendrocytes in the white matter [36,37,38], targets of research on cerebral white matter disorders, are also now considered to be important components of the NVU [19,20,21] (Figure 1).

## 3. Current Status of Research on the NVU and the Positioning of Astrocytes

When the concept of the NVU was first conceived, the clinical benefits of pharmacological thrombolysis with alteplase, a recombinant tissue plasminogen activator (rt-PA), for recanalization/reperfusion therapy for patients with acute ischemic stroke had already been established [39,40]. The concept of NVU was developed with the aim of developing complementary neuroprotection therapy to recanalization/reperfusion for patients with acute ischemic stroke [6,15,16,17,18]. Neuroprotection was expected to be realized by the discovery and clinical application of key molecules to potentially delay or halt neuronal cell damage that progresses in the interval of time that it takes for the therapeutic intervention to improve the cerebral circulation [41,42]. Among the many failures of animal to human applications, one achievement that showed potential for success was the development of the NA-1 peptide (nerinetide) by the group led by Michael Tymianski in Canada [43,44,45,46,47]. In the pathway of glutamate toxicity, the axis of the N-methyl-D-aspartate (NMDA) receptor, postsynaptic density protein 95 (PSD-95), and neuronal nitric oxide synthase (nNOS) signaling is central to ischemic neuronal damage, and blockade of this axis by NA-1 peptide was demonstrated to have the potential to offer neuroprotection in clinical trials for acute ischemic stroke in humans [43,44,45,46,47].

Over the past two decades, the need to understand the temporal and spatial diversity of the protective and injurious effects of the components of the NVU has also become clear. In particular, astrocytes have been shown to provide ever-changing metabolic support to neurons under physiological conditions [19,20,21]. However, they may exert not only protective, but also injurious effects due to their own ischemic dysfunction [19,20,21,41,42], making it difficult to establish therapeutic strategies, which was the original purpose of introducing the concept of the NVU [6]. The concept of the NVU was proposed with the goal of devising measures for “neuronal protection” based on a precise understanding of the mechanisms of neuronal injury. The goal has now shifted from “neuroprotection” to “whole NVU protection” (in other words, cerebral “cytoprotection” or “cerebroprotection”) [41,42]. In this context, the importance of astrocytes in the NVU is gaining more attention at present, because these cells play a neuroprotective role in conjunction with the microglia [32,33,34,35] and oligodendrocytes [36,37,38]. In particular, the influence of “help-me signals” emitted from vulnerable neurons in the early stages of ischemia on the astrocytes has attracted a lot of attention in recent years [41,42].

## 4. Anatomical Features of Astrocytes in the NVU

Neurons, which play a central role in brain activity and are considered to be the most vulnerable to ischemia, require an uninterrupted supply of oxygen and glucose via the CBF to satisfy their energy needs, and any interruption of this supply could lead to rapid functional decline and irreversible brain damage [15,16,17,18,41,42]. Thus, ischemic neuronal damage depends on the anatomically defined spatial arrangement of the neurons and microvessels; in Baboon’s histological examination of the brain, the mean distance between the cell bodies of striatal neurons and the closest microvessels (from arterioles to capillaries; less than 100 μm in diameter) was approximately 30 μm. During ischemia, individual analysis of the degree of neuronal damage shows that damage begins in areas beyond this average distance [48,49]. Therefore, the anatomical scale of the NVU, perfused by microvessels, is considered to be on the order of about 30 μm. Of note, the density of blood vessels is significantly lower in the cerebral white matter than in the cerebral cortex, and the physical size of the NVU centered on the blood vessels is not the same in all regions of the brain [48,49]. Furthermore, in the cerebral white matter, the NVU does not include the cell bodies of neurons, but axons and oligodendrocytes; the ischemic response here is also different, so NVUs in the grey and white matter cannot be treated as being qualitatively the same [19,20,21].

Astrocytes are always present between neurons and capillaries. Astrocytic foot processes (end feet) are a crucial component of the NVU, since they are interposed between the vascular and neuronal (synaptic) sides of the brain and make contact with three components of the NVU [50]. First, they envelop the synapse, forming a tripartite synapse; second, they make contact with the cerebral microvessels and capillaries, where the astrocyte foot processes encircle the microvessels [19,20,21]; third, astrocytes connect with each other through gap junctions [51], which are thought to allow direct exchange of micro-materials (K^+^, Ca^2+^, lactate, etc.) of less than 1.2 kDa between astrocytes, resulting in the formation of a single syncytium that may be involved in the transmission of tissue damage signals between neighboring NVUs and expansion of the damage area [52]. It is easy to imagine that such an anatomical structure would have significant implications for the responses of the NVU to cerebral ischemia [15,16,17,18,41,42]. Glucose and oxygen delivery from intravascular sources may be transported to neurons via astrocytes and the regulation of CBF for neuronal activity may also be mediated by astrocytes. In fact, astrocyte-dependent vascular control has been shown to be mediated by a variety of factors and is one of the important functions of the normal NVU [4,8].

## 5. Regulation of Cerebral Blood Flow by Astrocytes in the NVU

It is evident from the fact that astrocytes cover 99% of the capillary surface, and are the site of first contact with the energy substrate supplied by the capillaries, that astrocytes are involved in the regulation of the CBF at the capillary level [4,8,53]. No glial cells, including astrocytes, generate any action potentials [19,20,21]. The functional activity (action potential) of the brain enhances its ATP production to replenish the ATP consumed by the Na^+^,K^+^-ATPase during the process of restoring the ionic concentration gradient of the neurons in the region, which entails an increase in the local cerebral metabolic rate of glucose (CMR_glc_) and the local cerebral metabolic rate of oxygen (CMRO_2_). To allow for this increase, the local CBF also increases [4,8]. In fact, the local CMR_glc_ rather than the local CMRO_2_ reflects the neural activity better, because of the time lag to changes in the CMRO_2_. Indeed, it has also been established that there is a quantitative linear proportional relationship between the local CBF and CMR_glc_ [22,23].

The glucose content in the cerebral blood is 6 mmol/L, which differs from the oxygen content of 8–9 mmol/L. Complete oxidation of glucose requires 6 moles of oxygen per mole of glucose, and 30% of the oxygen and 10% of the glucose in the cerebral arterial blood are taken up and consumed [22,23]. In other words, there is a relative surplus of glucose supply. Furthermore, in the ischemic core of cerebral infarction (penumbra area), the glucose supply exceeds the oxygen supply due to the presence of residual CBF, creating an environment in which anaerobic glucose metabolism is likely to occur [19,20,21]. Conversely, when the CBF increases, the amount of glucose and oxygen supplied greatly exceeds the increase in the local CMR_glc_ and CMRO_2_. Therefore, it has been proposed that the increase in local CBF is for the purpose of heat disposal rather than for the supply of energy substrates. It is generally true that functional hyperactivity and increased CBF are tightly linked (functional hyperemia) [4,8], and also linked to increased glucose consumption [22,23]. This is one of the most fundamental features of the NVU. Functional hyperemia was first reported in the 1800s by Mosso [54], and later confirmed by Roy and Sherrington [55]. Later, a quantitative correlation between the local CBF and local CMR_glc_ in the brain was confirmed by Sokoloff and Kety [56]. Thus, functional hyperemia can be referred to as flow–metabolism coupling. An important question is whether CBF regulation can actually occur at the capillary level and, if so, how are the astrocyte foot processes involved in it? In the next section, I will review these issues discussing the possible role of astrocytes in CBF regulation, which was also reviewed in the previous review article [20].

## 6. Group I Metabolic Glutamate Receptors in the Astrocytes

The paper published by Zonta et al. in 2003 [57,58] reported a major breakthrough, both in relation to the mechanisms of functional hyperemia and the functions of the astrocytes. Using cortical slice preparations from relatively young rats (9–15 days of age), they linked changes in the cerebral arterial caliber to changes in the intracellular Ca^2+^ concentrations in the astrocytes. They found that the release of glutamate, the most ubiquitous excitatory neurotransmitter in the brain, initiates a response by stimulating the metabotropic glutamate receptors (mGluRs) expressed in the astrocytes. Among the several types of mGluRs expressed in astrocytes, group I mGluRs, which are conjugated to phospholipase C (PLC)/inositol-1,4,5-triphosphate (IP3), are important, and the increase in IP3 is associated with intracellular IP3-sensitive Ca^2+^ stores. Ca^2+^ release from the endoplasmic reticulum (ER) induces an increase in the intracellular Ca^2+^ concentration. Subsequent activation of Ca^2+^-sensitive phospholipase A2 (PLA2) releases arachidonic acid (AA) from the plasma membrane, which, in turn, releases prostanoids produced from AA via cyclooxygenases (COXs); in particular, prostaglandin E2, which is produced from AA via COX, relaxes the smooth muscle of the cerebral arterioles. Astrocytes are known to express Na^+^-dependent glutamate transporters (GLT-1 and GLAST), which rapidly recover the glutamate released between synapses and serve as stimulatory signals from the neurons for astrocyte glucose metabolism, and this study highlighted the actions of the mGluRs.

## 7. Epoxyeicosatrienoic Acids

Harder et al. [59] focused on the cerebral vasodilatory response to epoxyeicosatrienoic acids (EETs) released from the astrocytes. They hypothesized that AA is released from the plasma membrane, not only by the increase in intracellular Ca^2+^ and Ca^2+^-sensitive PLA2 induced by mGluR stimulation, but also by diacylglycerol (DAG)-activated phospholipase activated by PLC-induced increase in DAG. CYP2C11, which has been cloned as a cytochrome P450 (CYP) enzyme in astrocytes, exerts epoxygenase activity and catalyzes the synthesis of EETs from AA. Among these, 11,12-EETs and 14,15-EETs are reported to represent the main body of stable and potent vasodilators. In addition, EETs have been reported to be one of the endothelium-derived hyperpolarizing factors (EDHFs) that cause vasodilatation. In addition to de novo synthesis, EETs in storage form in the astrocyte membrane are also known to be released in response to intracellular Ca^2+^ increases in astrocytes. As described here and in Section 6, it is of note that at least two laboratories focused on astrocytic involvement in the regulation of CBF approximately 20 years ago.

## 8. Constriction of the Cerebral Microvessels by Astrocytes

At the same time as Zonta et al., a completely opposite phenomenon of vasoconstriction induced by astrocytes was reported by Mulligan and MacVicar [60]. They used 13- to 18-day-old rats or mouse hippocampal slices to increase the intracellular Ca^2+^ concentration in the astrocytes by photolysis of caged Ca^2+^, to promote AA release from the astrocyte cell membranes. They concluded that AA is converted to 20-hydroxyeicosatetraenoic acid (20-HETE) by ω-hydroxylation catalyzed by CYP 4A enzymes in the vascular smooth muscle, which causes vasoconstriction. Interestingly, in their experiments, treatment of the hippocampal slices with noradrenaline induced vasoconstriction by a similar mechanism, suggesting that noradrenergic neurons, which are derived from the nucleus accumbens in the brain parenchyma and are representative of intrinsic pathways, are responsible for vasoconstriction in the brain parenchyma. In fact, the vasoconstrictor effect of noradrenaline in the brain parenchyma is mediated by astrocytes. In fact, astrocytes encircling the vessel walls contain many nerve endings of noradrenergic neurons.

Subsequently, Gordon et al. [61,62,63] reported that astrocytes can induce both vasodilatation and vasoconstriction in response to the state of oxygenation. Astrocytes avidly produce lactate, which is exaggerated under the hypoxic condition, and an increase in extracellular lactate levels leads to the inhibition of prostaglandin E2 recovery by the astrocytic prostaglandin transporters. Therefore, prostaglandin E2-mediated vasodilatation is enhanced under the hypoxic condition, while 20-HETE causes vasoconstriction under the condition of oxygen abundance. In fact, lactate production in astrocytes and its extracellular release are the most important aspects of astrocyte metabolism in the NVU [19,20,21].

## 9. Functional Hyperemia at the Capillary Level

Previous reports have indicated that vasoactive substances released from the astrocyte foot processes may regulate the CBF via contraction and dilatation of the vascular smooth muscle at the level of the brain microvessels. However, once again, it is at the capillary level that the astrocyte foot processes have the greatest influence [4,8]. The question of whether the capillaries themselves are responsible for the autonomous increase or decrease in the size of the vascular bed draws attention to the function of the pericytes that reside here [24,25,26,27,28]. It has long been reported that pericytes contain actin filaments, which have a contractile effect. However, even if pericytes existed at the capillary level immediately after the bifurcation of small arteries and surround the vessels, and cell contraction immediately reduces the vessel caliber, it remains controversial if these cells can be considered pericytes or whether they should be considered as a part of the vascular smooth muscle. If PDGFβ-positive cells that exist in the cerebral blood vessels, which are at least clearly seen at the capillary level, are defined as pericytes, the point of debate is whether these cells function to regulate the size of the capillary bed. Recent studies have reported that pericytes at the capillary level do indeed have a regulatory effect on the CBF, but they are only responsible for very slow and gradual changes of the blood flow [24,25,26,27,28].

Another hypothesis suggests that astrocytes directly regulate the size of the vascular bed at the capillary level [29,30,31]. Aquaporin 4 (AQP4), which is strongly expressed in the vascular peduncle of astrocytes, is a water channel that regulates the movement of water molecules in and out of the cell; AQP4 channels are key molecules that induce cellular edema in the event of ischemic neuronal injury, but they also regulate the water content in the intracellular compartment within the astrocytes and extracellular compartment, or Virchow–Robin space, under physiological conditions. These channels monitor the metabolic state of astrocytes and close when the intracellular H^+^ increases due to metabolic activation, thereby inhibiting water movement out of the cells and narrowing the Virchow–Robin space [29,30,31]. This immediately dilates the capillary bed, potentially increasing the blood flow. Acidification in astrocytes is caused by various metabolic changes, and is particularly related to glucose metabolism. The astrocyte–neuron lactate shuttle (ANLS) model (Figure 2) [19,20,21], which will be discussed in the following section, lends support to the possibility that the lactate produced in astrocytes may cause acidification. An alternative pathway for lactate production may be glutamate taken up by the neurons in the glutamate–glutamine cycle. After conversion to α-ketoglutarate, glutamate undergoes metabolism in the tricarboxylic acid (TCA) cycle and may be partly involved in ATP production in the astrocytes and a source of lactate production by malic enzyme after metabolism in the TCA cycle [19,20,21].

In general, astrocyte regulation of blood flow is an attractive proposition, but at present, it seems to be controversial. Some data support that the vascular smooth muscle at the level of the small arteries/arterioles is primarily responsible for functional hyperemia and the rapid increase in blood flow in response to neural activity. Moreover, it is not the astrocytes, but rather the hyperactive neurons themselves that are thought to exert the major influence here [4,8].

## 10. Glymphatic System and Cerebral Circulation and Metabolism

Cerebral capillaries, the site of metabolism, along with the supply of oxygen and glucose, is also the site of metabolic product disposal [64,65,66,67,68,69,70,71]. Specifically, it is easy to conceive that astrocytes in contact with the capillaries utilize the Virchow–Robin space, their outer lumen, for disposal. Of particular interest is the glymphatic system (GS) as an excretory system for amyloid-β and phosphorylated tau proteins; it is well known that impaired activity of the GS leads to increased amyloid-β production and extracellular accumulation in the brain, and it is well known that, eventually, intracellular phosphorylated tau accumulation leads to neuronal death and Alzheimer’s disease [64,65,66,67,68,69,70,71]. It is not yet clearly understood if a distinction can be made between substances specifically excreted via the GS and those supplied and excreted by the normal arteriovenous system.

## 11. Glucose Metabolic Compartmentalization Formed by Cell Types in the NVU

So far, we have looked at glucose metabolism at the macro level in the whole brain, but it is thought that there are differences in glucose metabolism at the microscopic level in each cell type in the brain [19,20,21]. As already mentioned, the brain depends on the oxidative metabolism of glucose, but the measured ratio of CMR_glc_ to CMRO_2_ (CMRO_2_/CMR_glc_) is 5.5, which is slightly lower than the theoretical value of 6 that is obtained when glucose, a 6-carbohydrate sugar, is assumed to be completely oxidatively metabolized. In the late 1980s, in vivo PET conducted to investigate human brain metabolism showed that the CMRO_2_ and CMR_glc_ do not increase in tandem when the brain is functionally active, and that the latter increases to a greater degree than the former [72,73]. As a result, the CMRO_2_/CMR_glc_ decreased from 5.5 to 5.0 when the functional activity of the brain increased. This appears to be incompatible with the increase in the regional brain lactate production measured by magnetic resonance spectroscopy (MRS) [74], and raises the question of whether brain energy metabolism can be assumed to be based on oxidized phosphate in the mitochondria alone. There are two important points that should be borne in mind while interpreting this phenomenon: first, because hyperactivity of neurons is not a steady state, but a transitional state, its measurement involves a dynamic component that changes with time. In animal experiments, a decrease in CMRO_2_/CMR_glc_ was observed during beard stimulation in rats, subsequently followed by an increase. It appears certain that glucose consumption exceeds oxygen consumption at least at the very beginning of functional activity, leading to a transitory shift to a glycolytic-dominant metabolism. In 1994, Pellerin and Magistretti [75] proposed the ANLS model, which was tested and confirmed by the author [76] and others [77,78,79], to identify the cell component that showed a glycolytic-dominant metabolism. As tested and confirmed by the authors, astrocytes are the main source of lactate production. However, this hypothesis still remains to be verified [80,81,82,83,84,85,86] to this day, and is controversial [87,88,89,90,91,92,93,94,95,96,97]. However, if lactate undergoes oxidative metabolism completely in the mitochondria after a certain period of time, the theoretical value of CMRO_2_/CMR_glc_ would be 6. However, the actual CMRO_2_/CMR_glc_ is even lower than the value measured at rest, suggesting that some lactate may be discarded without undergoing oxidative consumption. This indicates that there may be some other roles of the enhanced glycolytic system [19,20,21].

## 12. Astrocyte–Neuron Lactate Shuttle Model

Even if the brain, as a whole, is dependent on the oxidative metabolism of glucose, the possibility of differences depending on the cell types still remains. Glucose is metabolized in the glycolytic system producing pyruvate, and pyruvate can be converted to lactate which, in turn, is transported to other cells. Lactate undergoes oxidative metabolism in the TCA cycle, resulting in oxidative glucose metabolism in the brain as a whole. The major difference between the glycolytic system and the TCA cycle is the difference in the efficiency of ATP production. The net production of ATP in the glycolytic system is 2 molecules of ATP per molecule of glucose, whereas the total ATP production through the TCA cycle reaches 32 molecules [23]. The main reason the brain needs such a large amount of ATP production is for maintenance of the ion concentration gradient by Na^+^,K^+^-ATPase. ATP consumption is particularly significant in cells that generate action potentials, i.e., neurons. Since astrocytes do not generate action potentials, the maintenance of ion concentration gradients in these cells has a different purpose [19,20,21]. Glutamate, a pervasive excitatory synaptic neurotransmitter in the brain, is released into the synaptic clefts, and is rapidly retrieved by the foot processes of the astrocytes that encircle the synapse. The astrocyte foot processes contain a Na^+^-dependent glutamate transporter, and glutamate is taken up by the astrocytes following an extracellular to intracellular Na^+^ concentration gradient. Whether the ATP needed for this is produced in the glycolytic system or the TCA cycle is an important question. Pellerin and Magistretti [75] observed that glutamate loading of cultured astrocytes resulted in increased glucose consumption and increased lactate release from astrocytes, reflecting a transient increase in glycolytic activity during in vivo neuronal hyperactivity. Since glucose uptake and consumption also occur in neurons, glucose is supplied by the blood vessels and is first taken up by the astrocytes that are in contact with them, or diffuses through the extracellular fluid, to be taken up directly by the neurons. In vivo, the ratio of glucose uptake by the neurons and astrocytes is 1:1 to 1:2, and it appears to be beyond doubt that astrocytes consume more glucose than neurons. Astrocytes were originally thought to mainly show glycolytic metabolism. Although ATP production is highly efficient in the TCA cycle, this is not necessarily a glucose-derived metabolic pathway. In astrocytes, the glutamate taken up from the synapses is converted to glutamine and recycled back to neurons (glutamate–glutamine cycle), but it is also converted to α-ketoglutarate, which flows directly into the TCA cycle to produce ATP. One of the biggest questions raised by the ANLS model is whether the increased release of lactate from astrocytes results in a corresponding increase in its uptake by neurons. Lactate uptake into neurons occurs by concentration-dependent passive diffusion, aided by monocarboxylate transporters (MCTs) in the plasma membrane, but MCT2 expressed in neurons is already saturated with low concentrations of lactate, because it is a low-K_m_ high-affinity MCT. It has been argued that uptake in the ANLS model does not increase even when the extracellular lactate concentration increases from the normal lactate concentration in the brain of 1–3 mM. On the other hand, the most important point of the ANLS model is that it raises the question of why astrocytes metabolize glucose via the glycolytic system.

## 13. Link between Glucose Metabolism, Lipid Metabolism and Ketone Bodies

Another aspect of astrocyte glucose metabolism to consider is fatty acid metabolism and ketone body synthesis [19,20,21,98,99]. Ketone bodies are usually produced in the liver during impaired glucose utilization. β-hydroxybutyrate and acetoacetate, which are the result of β-oxidation of medium-chain fatty acids, are representative ketone bodies. Ketone bodies are a source of energy for the adult brain only during starvation and in the pathological setting of insulin resistance. The transport of ketone bodies from the blood to the brain is carried out by MCTs, which, like lactate, are essentially transported from the blood to the astrocytes as well as neurons via the endothelium. In addition to these exogenous ketone bodies, astrocytes themselves have the ability to synthesize endogenous ketone bodies. The regulation of ketone body synthesis in astrocytes is controlled by the intracellular adenosine monophosphate (AMP) concentration, which activates AMP-activated protein kinase (AMPK) when the intracellular energy state is low. In other words, it has been shown that ketone body synthesis is enhanced when astrocytes are exposed to a hypoxic and low-glucose environment [100]. Ketone bodies are converted to acetyl CoA, which serves as the substrate for the TCA cycle. This point is similar to that for lactate, but the critical difference is the necessity of the pyruvate dehydrogenase complex (PDHC) for pyruvate/lactate metabolism. Both lactate and pyruvate are converted to acetyl CoA by PDHC, whereas conversion of ketone bodies to acetyl CoA does not need PDHC action. During ischemia–reperfusion, PDHC is susceptible to damage by reactive oxygen species. As a result, the accumulated lactate under the hypoxic condition remains restricted for use by neurons after reoxygenation. However, ATP production from ketone bodies in the TCA cycle can resume immediately in the presence of oxygen. In addition, ketone bodies have long been known to exert a neuroprotective effect [19,20,21,98,99,100]. Furthermore, ketone bodies are a biosynthetic substrate for cholesterol, which is a component of the myelin sheath. Myelin, or oligodendrocytes, which are highly sensitive to hypoxia, are susceptible to ischemic injury. In white matter that has escaped axonal injury, remyelination is essential for functional recovery, and potential use of ketone bodies for remyelination in the living brain, where the presence of oligodendrocyte precursor cells (OPCs) has been confirmed, has attracted attention [19,20,21,98,99,100].

## 14. D-Serine and L-Serine

D-serine (DS) is essential for glutamatergic neuronal excitation as an NMDA receptor co-agonist, but exacerbates neurological damage during cerebral ischemia [101,102]. LS is metabolized from L-serine (LS) by serine racemase (SRR) in neurons, and synthesized by 3-phosphoglyceratedehydrogenase (3PGDH) in astrocytes [101,102]. The metabolic regulation of DS and LS in the ischemic foci of the brain has not yet been fully elucidated, although our group also investigated it using an in vitro rat cell culture system [102]: cultured neurons and astroglia were prepared from SD rats, and the changes in the concentrations of DS and LS in the medium after 24 h of hypoxia in a 1% O_2_ chamber were measured by two-dimensional HPLC. The net production rate was quantified. The concentration of DS in the supernatant of the cultured neurons was significantly increased (*p* < 0.001) in the neurons exposed to hypoxia as compared with that in the control cells, but no such change was observed in the astroglia; conversely, the concentration of LS was significantly increased (*p* < 0.05) in the astroglia, while no such change was observed in the neurons. It is thought that the release of DS from neurons and LS from astrocytes increases during cerebral ischemia [102].

As an in vivo model of cerebral ischemia in mice, an embolization thread made of nylon thread was inserted into the internal carotid artery of wild-type (WT) and SRR-/- mice, until the origin of the middle cerebral artery (MCA) and the MCA was occluded. There was no significant difference in the degree of CBF reduction after MCA occlusion between the WT and SRR-/- groups, but the infarct volume at 24 h after reperfusion was significantly (*p* < 0.05) smaller in the SRR-/- mice. Although the DS content peaked at about 20 h in the post-ischemic tissues, the concentration of DS in the SRR-/- mice was about one-tenth that in the WT mice, suggesting that the reduced infarct volume in the former mice was probably associated with the reduced DS content. Immunostaining showed that the expression of SRR was enhanced in the neurons on the ischemic side, suggesting that neurons are the source of DS during cerebral ischemia [102]. Interestingly, the increase in the LS content in the ischemic foci was much greater than that of the DS content in the WT mice, suggesting that the significance of LS as a cytotoxic factor may require further investigation. Furthermore, LS has also been reported to have a cytoprotective effect [103,104,105]. The fact that 3PGDH, which is expressed specifically in astroglia, is branched off from the glycolytic system to synthesize LS and that the glycolytic system is enhanced under ischemia are suggestive of astrocytic neuroprotective roles through high glycolytic activity [19,20,21].

## 15. Mitochondrial Transfer of Astrocytes and Neurons

Recently, Hayakawa et al. reported observing the phenomenon of mitochondria being donated from astrocytes to neurons [106,107,108,109,110,111,112,113,114]. We have discussed the transfer of metabolites from astrocytes to neurons [19,20,21], which demonstrates how astrocytes create an efficient and viable environment in the metabolic system of neurons, whose primary goal is efficient production of ATP. In fact, mitochondria are constantly exposed to oxidative stress in the process of oxidative metabolism of glucose. Oxidative stress is one of the most important factors implicated in the pathogenesis of many neurodegenerative diseases, for which age-related mitochondrial dysfunction is the most important risk factor. The mechanism by which mitochondria are supplied from astrocytes to neurons is considered to represent the strongest compartmentalization in the NVU. At present, it has been confirmed that mitochondria are activated after ischemic stroke, but the universality of this mechanism in degenerative diseases still needs to be examined.

Mitochondria transfer has been investigated in neurological diseases other than stroke models. In Parkinson’s disease, mitochondrial dysfunction involving oxidative stress, alpha-synuclein oligomers and aggregates plays a major role in the degeneration of dopaminergic neurons, and replacing the mitochondria of dysfunctional dopaminergic neurons with healthy mitochondria may have therapeutic effects. Mitochondria themselves in vitro have a short lifespan of only a few hours to a few days, so it is necessary to have a constant supply of cells in the brain in vivo. In a co-culture system of human induced pluripotent stem (iPS) cell-derived dopaminergic neurons and astrocytes, Cheng and colleagues [115] showed that astrocyte-derived mitochondrial transitions occur and restore the function of rotenone-injured dopaminergic neurons. It has been suggested that the phospho-p38-dependent pathway is involved in the mitochondrial transition, which is not a CD38-dependent manner [108,109].

It is noteworthy that mitochondrial transfer between astrocytes and neurons also occurs from neurons to astrocytes, and that it also occurs between astrocytes and oligodendrocytes. Alexander’s disease, caused by mutations in glial fibrillary acidic protein (GFAP), which forms the cytoskeleton of astrocytes, is a leukoencephalopathy resulting in cerebral white matter lesions due to abnormalities in the vascular endothelium and oligodendrocytes. Gao et al. [116] found that mitochondrial transition was impaired in astrocytes derived from human iPS cells transduced with GFAP mutations. Again, a CD38-dependent pathway is involved, and these cell-to-cell mitochondrial transitions are expected to play a major role in elucidating the maintenance of NVU function in both cerebral white matter and gray matter, and in disease treatment strategies

## 16. Conclusions and Issues That Remain to Be Resolved

Until now, cultured rodent cells have been used for functional analyses of astrocytes. Based on the metabolic profile of rodent astrocytes, the neuroprotective aspects of these cells, such as their antioxidant effects, have been demonstrated. The metabolic profile of human astrocytes is still not well known [117,118], even though it has been believed to be similar to the metabolic profile in rodents. However, a recent paper has suggested the possibility that the metabolic profiles of astrocytes in rodents and humans may not be similar at all [117]. In other words, astrocytes isolated acutely from humans may not necessarily exhibit high antioxidant activity. On the other hand, we have induced iPS cells to differentiate into astrocytes and spinal motor neurons and analyzed the metabolic profiles of these cells [118]. At present, we have confirmed that the pentose-phosphate pathway (PPP) flux in astrocytes is several times higher than that in neurons and does not differ significantly from that in rodent astrocytes. Further detailed analyses are required. It is also necessary to examine the detailed metabolic profile of human iPS cell-derived microglia, since methods to induce differentiation of iPS cells into microglia have been recently established, and to study their interactions with astrocytes.

## Figures and Tables

**Figure 1 cells-11-00813-f001:**
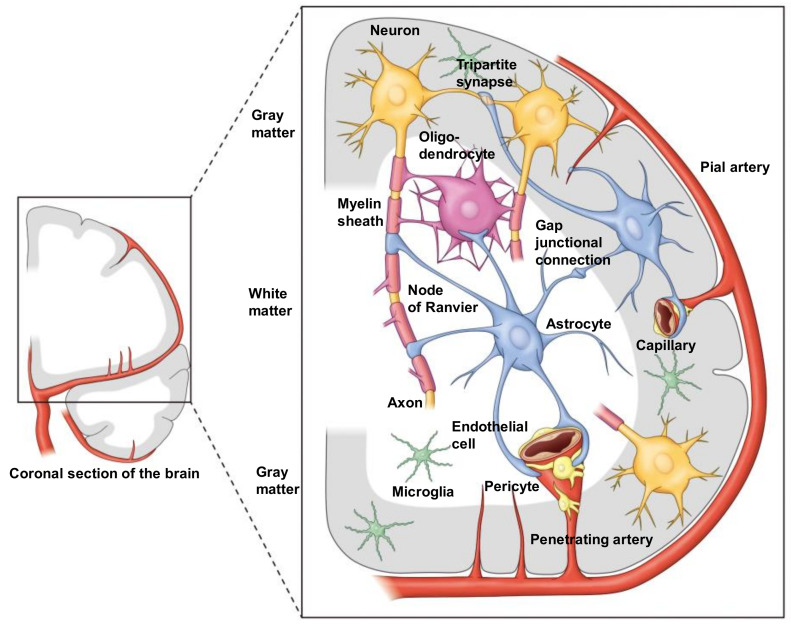
Color illustration depicting the “NVU” in both gray and white matter of the human brain. In white matter, both astrocytes and oligodendrocytes are involved in the NVU, since a major part of the neuronal axon is myelinated by oligodendrocytes. Importantly, however, astrocytes can contact axons directly at the site of the Ranvier node. Note that it has not been reported that oligodendrocytes have direct contact with microvessels, since astrocytic endfeet cover microvessels almost completely. Therefore, astrocytes contact oligodendrocytes as well as axons. One of the important roles of astrocyte is the regulation of synaptic transmission. The endfeet of astrocytes envelope synapses (i.e., tripartite synapse). Moreover, astrocytes connect themselves with connexin gap junctions, forming a huge syncytium. At present, the anatomical location of microglia in the NVU is not specified.

**Figure 2 cells-11-00813-f002:**
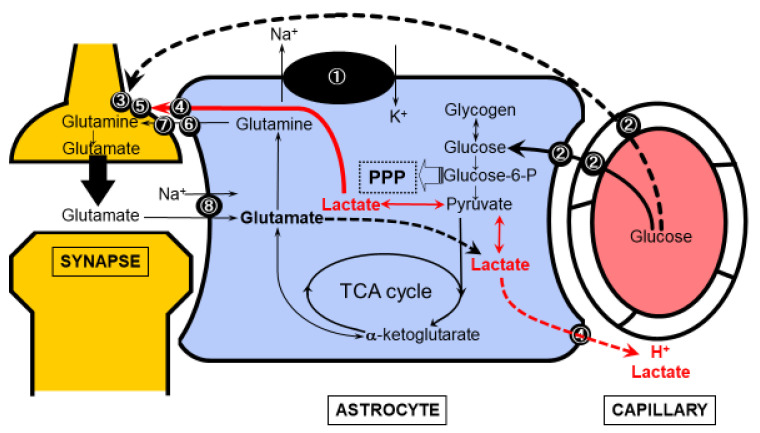
The astrocyte-neuron lactate shuttle (ANLS) model. Glucose is supplied from within the brain capillaries to neurons and astrocytes. For neurons, in addition to the direct supply shown by the dotted line, glucose-derived lactate metabolized in astrocytes may be supplied to neurons as an energy substrate. Importantly, lactate derived from either glucose or glutamate in astrocytes can act as vasodilatory signal on capillaries in the brain. (adapted from [20]). ① Na^+^,K^+^-ATPase ② Glucose transporter 1 (Glut1) ③ Glucose transporter 3 (Glut3) ④ Monocarboxylic acid transporter (MCT) 1&4 astrocytic form ⑤ Monocarboxylic acid transporter (MCT) 2 (neuronal form) ⑥ System N transporter (astrocytic form) ⑦ System A transporter (neuronal form) ⑧ Na^+^-dependent glutamate transporter (GLT1, GLAST).

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
