# Peer review of "Metabolic Contribution and Cerebral Blood Flow Regulation by Astrocytes in the Neurovascular Unit"

_cells, 2022, doi:10.3390/cells11050813_

Round 1

Reviewer 1 Report

Most of the information enclosed in the manuscript is already published.

Previous publication:

Metabolic compartmentalization between astroglia and neurons in physiological and pathophysiological conditions of the neurovascular unit 

Shinichi Takahashi 1 2 

Affiliations 

  • 1Department of Neurology and Stroke, Saitama Medical University International Medical Center, Saitama, Japan. 
  • 2Department of Physiology, Keio University School of Medicine, Tokyo, Japan. expand 
  • PMID: 32037635  
  • PMCID: PMC7187297  
  • DOI: 10.1111/neup.12639 

-----------------------------------------------------------------------------------

Review: commenting on the completeness of the review topic covered, the relevance of the review topic, the gap in knowledge identified, the appropriateness of references, etc.

The publication is a continuation of an author's earlier work (DOI: 10.1111/neup.12639) and is supposed to supplement the information on the neurovascular unit.

Specific comments refer to line numbers, tables, or figures that point out inaccuracies within the text or sentences that are unclear. These comments should also focus on the scientific content and not on spelling, formatting, or English language problems, as these can be addressed at a later stage by our internal staff.

  1. In the draft are many repeated thoughts and statements that extend the main text, without giving new information. e.g. Line 101 – 103, line 131-135, line 148-150, line 160-164, line179-171 vs introduction.
  2. Figure 1 and the information of metabolic compartmentation between cells, were already reviewed in the previous author's publication (DOI: 10.1111/neup.12639), but this is not marked neither in main text nor in Figure 1, as self-citation. Such a situation can indicate the self-plagiarism.
  3. Figure 2: required measurement units are missing.
  4. Figures 1, 2, and 3 – without a citation and there is no mention about the authorship of the presented data.
  5. Information from Chapter 5 (lines 165 – 197) was referred earlier (DOI: 10.1111/neup.12639), which can be treated as self-plagiarism.
  6. Chapter 7 (“Epoxyeicosatrienoic acids”) seems to be in draft form, where the citations have the wrong format. Moreover, the information enclosed here is incomplete, unclear, and out of context.
  7. Typo: chapter number (line 198); citation line 207; citation line line 221; citation line 229;
  8. Improper vocabulary: vasodilatation instead vasodilation (Chapter 8).
  9. Sentence lines 293 – 296 stay in contradiction to the rest information presented in the article but the discussion is not extended to further analysis (Chapter 9).
  10. A huge part of the text refers to the dictionary or coursebook knowledge rather than being a valuable summary of the latest scientific publications. E.g Chapter 11 (line 325 – 361) refer to the kinetic aspects of glycolysis rate and hexokinase activity. This information can be read in each biochemical book. Moreover, the author incorporates their improper self-citation there [22].
  11. Missing sentence part: line 399.
  12. Part of the information is incorrect, based on too old studies (where the content of which was later verified in newer scientific studies). E.g. “ line 404 – 405. “The net production of ATP in the glycolytic system is 2 molecules of ATP per molecule of glucose, whereas, in the TCA cycle, it is 36 molecules.”
  13. Chapter 15 - unclear and out of article context.
  14. They are a number of previous review articles that referred better and more comprehensively to the ANLS model.
  15. Interesting chapters about new scientific subjects are marginally discussed: Chapter 10 (“Glymphatic system and cerebral circulation and metabolism”) and Chapter 16 ("Mitochondrial transfer of astrocytes and neurons").
  16. Some citations are from an abstract conference book.

For many reasons (points 1-16), this publication does not make a valuable contribution in the referred field.

Author Response

Summary of Responses for Editorial Manager and Reviewers

  1. Title: the title of this review article has been modified from “Metabolic contribution of astrocytes in the neurovascular unit” in the original manuscript to “Metabolic contribution and cerebral blood flow regulation by astrocytes in the neurovascular unit” in the revised version. The contents of this article focus on dual roles (i.e., “Metabolic contribution” and “CBF regulation”) of astrocytes in the NVU. In fact, the reviewer #1 has criticized that some parts of the manuscript are not relevant. Therefore, the author has changed the title of the revised version to make a purpose of this review article clearer.

  1. Reference #6: this reference is a conference proceeding and has not been published as original article. However, the whole contents, which are available on the indicated URL (https://www.stroke.nih.gov/documents/SPRG_report_042002_508C.pdf), are extremely important. Thus, it should be listed in the references. Unfortunately, however, the URL of reference #6 was handled as a different reference (#7) after submission (just because of “new line”), and it seemed that there are two conference papers listed (Refs #6 and #7). I have fixed this error and reference #7 disappeared (In the very original submission, “#7” was used for the next paper). Importantly, this procedure has not affected the rest of the references. The numbers cited in the original text are exactly the same and correct as described in the text, and not changed in the revised version.

  1. Figures: A new figure (a color illustration) has been created and inserted before “original Fig. 1” according to the suggestion by Reviewer #2. As for “original Fig.1 (now, Fig. 2)”, the cells and vessels have been colored according to color used in the new Fig. 1. As for the original Figs. 2 and 3, I have deleted both figures to respond the comment by Reviewer #1 (see below). 

  • Specific response to Reviewer #1

Review: commenting on the completeness of the review topic covered, the relevance of the review topic, the gap in knowledge identified, the appropriateness of references, etc.

The publication is a continuation of an author's earlier work (DOI: 10.1111/neup.12639) and is supposed to supplement the information on the neurovascular unit.

Specific comments refer to line numbers, tables, or figures that point out inaccuracies within the text or sentences that are unclear. These comments should also focus on the scientific content and not on spelling, formatting, or English language problems, as these can be addressed at a later stage by our internal staff.

  1. In the draft are many repeated thoughts and statements that extend the main text, without giving new information. e.g. Line 101 – 103, line 131-135, line 148-150, line 160-164, line179-171 vs introduction.

Ans.----- Line 101 – 103, for “realization of neuroprotection in the light of NVU”; line 131-135, for “neuronal vulnerability due to anatomical location in the light of NVU”; line 148-150, for “astrocytic roles in neuroprotection in the light of NVU”; line 160-164, for “CBF regulation by astrocytes in NVU”; I do not think that these 4 parts are not merely repetitions of the introduction part of NVU, but its 4 specific detailed roles. Thus, I have left them untouched. (As for the 5th part, “line 179-171” does not make sense. )    

  1. Figure 1 and the information of metabolic compartmentation between cells, were already reviewed in the previous author's publication (DOI: 10.1111/neup.12639), but this is not marked neither in main text nor in Figure 1, as self-citation. Such a situation can indicate the self-plagiarism.

Ans.-----Although Fig. 1 looks like figures in the published article (DOI: 10.1111/neup.12639), it is clearly different from any figures there. In fact, the Fig. 1 emphasizes the pathway of lactate production and its effect on microvessles. Therefore, it is meaningful and necessary. However, this figure shares a similar concept of astrocytic metabolic contribution in the NVU, I have decided to cite this figure as “Figure 1 adapted from [20]” in the published article (DOI: 10.1111/neup.12639) ”.

  1. Figure 2: required measurement units are missing.

Ans.-----Taking the reviewer’s comments #3 as well as #10, I have decided to remove this figure.

  1. Figures 1, 2, and 3 – without a citation and there is no mention about the authorship of the presented data.

Ans.-----As for Fig.1 (now, “Fig. 2” in the revised manuscript, because new Fig. 1 has been created and inserted), I have added citation # as described in my response to comment #2. As for Figs. 2 and 3, I have decided to remove it because they are just textbook information and less relevant to this review article (see, Reviewer’s comment #10).

  1. Information from Chapter 5 (lines 165 – 197) was referred earlier (DOI: 10.1111/neup.12639), which can be treated as self-plagiarism.

Ans.-----This chapter provides fundamental information with regard to the cerebral blood flow and metabolism. Thus, I added one sentence in the end of this chapter citing the previous review article (DOI: 10.1111/neup.12639), which leads to help bridging the next chapter (Chapter 6).

  1. Chapter 7 (“Epoxyeicosatrienoic acids”) seems to be in draft form, where the citations have the wrong format. Moreover, the information enclosed here is incomplete, unclear, and out of context.

Ans.-----I have corrected the format of citation properly (deleted). I have added the conclusive comment to emphasize the importance of these works (Chapter 6 and 7)

  1. Typo: chapter number (line 198); citation line 207; citation line line 221; citation line 229;-----àI have corrected typos: chapter number (line 198); citation line 207; citation line line 221; citation line 229

  1. Improper vocabulary: vasodilatation instead vasodilation (Chapter 8).

Ans.-----Both “vasodilatation” and “vasodilation” are frequently used in the published literatures in this research field. I used “vasodilatation” throughout the text.

  1. Sentence lines 293 – 296 stay in contradiction to the rest information presented in the article but the discussion is not extended to further analysis (Chapter 9).

Ans.-----I have mentioned the contradiction and the controversy of this issue. In fact, these issues have been unsolved now. Thank you for valuable comments.

  1. A huge part of the text refers to the dictionary or coursebook knowledge rather than being a valuable summary of the latest scientific publications. E.g Chapter 11 (line 325 – 361) refer to the kinetic aspects of glycolysis rate and hexokinase activity. This information can be read in each biochemical book. Moreover, the author incorporates their improper self-citation there [22].

Ans.-----I decided to remove the whole chapter 11 since the information written in this chapter is not relevant to the NVU concept.

  1. Missing sentence part: line 399.

Ans.-----I have corrected the sentence to make it meaningful.

  1. Part of the information is incorrect, based on too old studies (where the content of which was later verified in newer scientific studies). E.g. “ line 404 – 405. “The net production of ATP in the glycolytic system is 2 molecules of ATP per molecule of glucose, whereas, in the TCA cycle, it is 36 molecules.”

Ans.-----I appreciate the reviewer’s comments pointing out the incorrect information in the light of recent studies. I rewrote this part citing the biochemistry textbook (already in the references).

  1. Chapter 15 - unclear and out of article context.

Ans.-----I do not agree with the reviewer since D-/L-amino acid compartmentation is relevant to the concept of metabolic contribution of astrocytes in NVU. I have left the whole chapter (now, “Chapter 14” in the revised version) as it is.

  1. They are a number of previous review articles that referred better and more comprehensively to the ANLS model.

Ans.-----I have chosen recent review articles despite numerous comprehensive review articles in the past.

  1. Interesting chapters about new scientific subjects are marginally discussed: Chapter 10 (“Glymphatic system and cerebral circulation and metabolism”) and Chapter 16 ("Mitochondrial transfer of astrocytes and neurons").

Ans.-----I have added 2 recent references to discuss these issues more deeply (following 2 paragraphs were added in Chapter 16 (now “15” due to deletion of Chapter 11)):

Mitochondria transfer has been investigated in neurological diseases other than stroke models. In Parkinson's disease, mitochondrial dysfunction involving oxidative stress, alpha-synuclein oligomers and aggregates plays a major role in the degeneration of dopaminergic neurons, and replacing the mitochondria of dysfunctional dopaminergic neurons with healthy mitochondria may have therapeutic effects. mitochondria themselves in vitro have a short lifespan of only a few hours to a few days, so it is necessary to have a constant supply of cells in the brain in vivo. In a co-culture system of human human induced pluripotent stem (iPS) cell-derived dopaminergic neurons and astrocytes, Cheng and colleagues [115] showed that astrocyte-derived mitochondrial transitions occur and restore the function of rotenone-injured dopaminergic neurons. It has been suggested that the phospho-p38 depended pathway is involved in the mitochondrial transition, which is not a CD38-dependent manner [108,109].

It is noteworthy that mitochondrial transfer between astrocytes and neurons also occurs from neurons to astrocytes, and that it also occurs between astrocytes and oligodendrocytes. Alexander's disease, caused by mutations in glial fibrillary acidic protein (GFAP), which forms the cytoskeleton of astrocytes, is a leukoencephalopathy resulting in cerebral white matter lesions due to abnormalities in the vascular endothelium and oligodendrocytes. Gao et al [116] found that mitochondrial transition was impaired in astrocytes derived from human iPS cells transduced with GFAP mutations. Again, a CD38-dependent pathway is involved, and these cell-to-cell mitochondrial transitions are expected to play a major role in elucidating the maintenance of NVU function in both cerebral white matter and gray matter, and in disease treatment strategies.

[115] Cheng XY, Biswas S, Li J, Mao CJ, Chechneva O, Chen J, Li K, Li J, Zhang JR, Liu CF, Deng WB: Human iPSCs derived astrocytes rescue rotenone-induced mitochondrial dysfunction and dopaminergic neurodegeneration in vitro by donating functional mitochondria. Transl Neurodegener. 2020 Apr 24;9(1):13.

[116] Gao L, Zhang Z, Lu J, Pei G: Mitochondria Are Dynamically Transferring Between Human Neural Cells and Alexander Disease-Associated GFAP Mutations Impair the Astrocytic Transfer. Front Cell Neurosci. 2019 Jul 10;13:316.

  1. Some citations are from an abstract conference book.

Ans.-----I believe that the Reviewer is talking about ref #6 and #7. As I have explained in the Summary Responses, Refs #6 and #7 are one conference proceeding (#6) and contains extremely important information. Thus, I have untouched this reference.

Reviewer 2 Report

Thank you for submitting your manuscript to this journal.

The manuscript is well thought out and organized with appropriate references.

Minor recommendations:  on line 67 ... in direct contact with neurons, neuronal may... consider replacing neuronal with neurons.

In Section 2. it would be good to place a nice image showing the structural arrangements of each cell of the NVU to assist your readers in addition to the word description within the text.  I strongly recommend the insertion of the image to strengthen you paper for the readers. 

Author Response

Summary of Responses for Editorial Manager and Reviewers

  1. Title: the title of this review article has been modified from “Metabolic contribution of astrocytes in the neurovascular unit” in the original manuscript to “Metabolic contribution and cerebral blood flow regulation by astrocytes in the neurovascular unit” in the revised version. The contents of this article focus on dual roles (i.e., “Metabolic contribution” and “CBF regulation”) of astrocytes in the NVU. In fact, the reviewer #1 has criticized that some parts of the manuscript are not relevant. Therefore, the author has changed the title of the revised version to make a purpose of this review article clearer.

  1. Reference #6: this reference is a conference proceeding and has not been published as original article. However, the whole contents, which are available on the indicated URL (https://www.stroke.nih.gov/documents/SPRG_report_042002_508C.pdf), are extremely important. Thus, it should be listed in the references. Unfortunately, however, the URL of reference #6 was handled as a different reference (#7) after submission (just because of “new line”), and it seemed that there are two conference papers listed (Refs #6 and #7). I have fixed this error and reference #7 disappeared (In the very original submission, “#7” was used for the next paper). Importantly, this procedure has not affected the rest of the references. The numbers cited in the original text are exactly the same and correct as described in the text, and not changed in the revised version.

  1. Figures: A new figure (a color illustration) has been created and inserted before “original Fig. 1” according to the suggestion by Reviewer #2. As for “original Fig.1 (now, Fig. 2)”, the cells and vessels have been colored according to color used in the new Fig. 1. As for the original Figs. 2 and 3, I have deleted both figures to respond the comment by Reviewer #1 (see below). 

  • Specific response to Reviewer #2

Comments and Suggestions for Authors

Thank you for submitting your manuscript to this journal.

The manuscript is well thought out and organized with appropriate references.

Minor recommendations:  on line 67 ... in direct contact with neurons, neuronal may... consider replacing neuronal with neurons.

Ans.―――>I have corrected the mistake according to the reviewer’s kind suggestion.

In Section 2. it would be good to place a nice image showing the structural arrangements of each cell of the NVU to assist your readers in addition to the word description within the text.  I strongly recommend the insertion of the image to strengthen you paper for the readers.

Ans.―――>I have added a color illustration depicting the “NVU” in both gray- and white-matter of the human brain. Especially emphasizing “NVU” in white matter, where both astrocytes and oligodendrocytes are involved in NVU, since most part of the neuron (axon) is myelinated by oligodendrocytes. In addition, astrocytes also contact axons directly at the site of Ranvier node. Importantly, it has not been reported that oligodendrocytes have a direct contact with microvessels. Astrocytic endfeet, which envelope microvessels almost completely, should contact oligodendrocytes in addition to axons.

This illustration was created by a professional company. Therefore, I have mentioned it in  Acknowledgments.

Round 2

Reviewer 1 Report

Accept in present form